# Different Impacts of DNA-PK and mTOR Kinase Inhibitors in Combination with Ionizing Radiation on HNSCC and Normal Tissue Cells

**DOI:** 10.3390/cells13040304

**Published:** 2024-02-06

**Authors:** Nina Klieber, Laura S. Hildebrand, Eva Faulhaber, Julia Symank, Nicole Häck, Annamaria Härtl, Rainer Fietkau, Luitpold V. Distel

**Affiliations:** 1Department of Radiation Oncology, University Hospital Erlangen, Friedrich-Alexander-Universität Erlangen-Nürnberg, Universitätsstr. 27, 91054 Erlangen, Germany; 2Comprehensive Cancer Center Erlangen-EMN (CCC ER-EMN), 91054 Erlangen, Germany

**Keywords:** AZD7648, cancer, CC-115, cell lines, HNSCC, DNA-PK, head and neck cancer, HNC: head and neck squamous cell carcinoma, ionizing radiation, kinase inhibitor, mTOR, PI3K, Sapanisertib, targeted therapy

## Abstract

Despite substantial advancements in understanding the pathomechanisms of head and neck squamous cell carcinoma (HNSCC), effective therapy remains challenging. The application of kinase inhibitors (KIs) in HNSCC, specifically mTOR and DNA-PK inhibitors, can increase radiosensitivity and therefore presents a promising strategy when used simultaneously with ionizing radiation (IR) in cancer treatment. Our study focused on the selective DNA-PK-inhibitor AZD7648; the selective mTOR-inhibitor Sapanisertib; and CC-115, a dual inhibitor targeting both mTOR and DNA-PK. The impact of these KIs on HNSCC and normal tissue cells was assessed using various analytical methods including cell death studies, cell cycle analysis, real-time microscopy, colony-forming assays and immunohistochemical staining for γH2AX and downstream mTOR protein p-S6. We detected a strong inhibition of IR-induced DNA double-strand break (DSB) repair, particularly in AZD7648-treated HNSCC, whereas normal tissue cells repaired DNA DSB more efficiently. Additionally, AZD7648 + IR treatment showed a synergistic decline in cell proliferation and clonogenicity, along with an elevated G2/M arrest and cell death in the majority of HNSCC cell lines. CC-115 + IR treatment led to an elevation in G2/M arrest, increased cell death, and a synergistic reduction in cell proliferation, though the effect was notably lower compared to the AZD7648 + IR- treated group. Sapanisertib led to a high cellular toxicity in both HNSCC and normal tissue cells, even in non-irradiated cells. Regarding cell proliferation and the induction of apoptosis and necrosis, Sapanisertib + IR was beneficial only in HPV^+^ HNSCC. Overall, this study highlights the potential of AZD7648 as a radiosensitizing agent in advanced-stage HPV-positive and negative HNSCC, offering a promising therapeutic strategy. However, the dual mTOR/DNA-PK-I CC-115 did not provide a distinct advantage over the use of selective KIs in our investigations, suggesting limited benefits for its application in KI + IR therapy. Notably, the selective mTOR-inhibitor Sapanisertib was only beneficial in HPV^+^ HNSCC and should not be applied in HPV^−^ cases.

## 1. Introduction

Head and neck squamous cell carcinoma (HNSCC) ranks amongst the most prevalent cancers globally, accounting for 890,000 new cases and 450,000 deaths annually. The incidence varies worldwide with concentrations in Europe, India, and Australia, and projections indicate a 30% annual increase by 2030 [1,2,3,4,5]. HNSCC risk factors are primarily tobacco and alcohol abuse, as well as infections with high-risk human papillomavirus (HPV) [2]. In clinical practice, HNSCC is commonly classified based on HPV status, as the tumor’s origin, oropharyngeal location, prognosis, and response to ionizing radiation (IR) depend on this distinction [6,7]. Particularly, HPV^−^ tumors are the most common locally advanced HNSCC subtype and have the poorest therapy response to both chemotherapy and IR, leading to a poor prognosis [8].

The majority of patients present with locally advanced-stage HNSCC and therefore require multimodal treatment approaches including surgery, anti-cancer drug therapy, and radiotherapy with IR [2]. Despite the rapid development of novel anticancer therapies, including monoclonal antibodies, as well as immune checkpoint inhibitors and small-molecule kinase inhibitors (KI), the long-term use of platinum-based cisplatin, which advanced in the 1970s, remains the primary first-line treatment for HNSCC in most cases [9,10,11,12,13,14,15,16]. An example for targeted therapy is Cetuximab, a monoclonal antibody targeting the epidermal growth factor receptor, which is FDA-approved for recurrent or metastatic HNSCC. In clinical trials, only ~10% of patients responded, while in the remaining patients a higher recurrence rate was observed [10,17,18,19,20]. Another FDA-approved example for treatment in recurrent or metastatic HNSCC is the immune checkpoint inhibitor Pembrolizumab, which binds to tumor cells expressing high levels of PD-L1 and achieved a response rate of ~20% in clinical trials; however, it led to hyperprogressive tumor disease in some patients [21,22]. Despite advancements in the knowledge on HNSCC epidemiology and pathogenesis, the prognosis of HNSCC remains poor, showing no significant improvement in survival rates and a high incidence of locoregional recurrence or distant metastases in more than half of HNSCC patients [2,23,24]. The critical demand for an improved understanding of therapy response and the development of new treatment options is particularly crucial for advanced-stage, HPV^−^ cases of HNSCC [25,26,27,28].

Recently, the combined administration of small-molecule kinase inhibitors and IR has been widely investigated. Particularly promising in this context are KIs that interfere with the cell cycle and DNA repair, as these two processes strongly influence and interact with radiotherapy [29]. IR induces DNA double-strand breaks (DSBs), which are considered the most deleterious form of DNA lesions. The primary modes of DSB repair are homologous recombination (HR) and non-homologous end-joining (NHEJ) [30,31,32]. Since HR is often impaired in cancer cells, they rely on NHEJ to survive [33]. Consequently, inhibiting NHEJ holds the potential for radiosensitizing cancer cells, resulting in cell death within irradiated cancer regions while sparing normal tissue [34,35,36].

A promising target protein for interfering with NHEJ is DNA-dependent protein kinase (DNA-PK), a member of the PI3K superordinate group within Phosphoinositide 3-kinase-related kinase (PIKK). The DNA-PK enzyme complex comprises the KU heterodimers KU70 and KU80 and a catalytic subunit DNA-PKcs [37,38,39]. It detects DSBs and initiates NHEJ [37,38,39]. Additionally, it plays a crucial role in important cellular processes, including the modulation of chromatin structure, telomere maintenance, signal transduction, and transcriptional regulation [40,41]. Therefore, inhibiting DNA-PK with KIs holds potential for improving the effectiveness of radiation therapy [42].

An alternative strategy for using the interaction between KIs and IR involves the modulation of the cell cycle. The synthesis phase of the cell cycle is the most IR-resistant, while the G1 and (particularly) G2/M phases are clearly more sensitive to IR exposure [43]. Within the spectrum of mutations identified in HNSCC, the phosphoinositide 3-kinase (PI3K)–AKT–mTOR pathway is the most frequently mutated (30.5%), and multiple mutations in this pathway always lead to a stage IV cancer [44]. The frequently mutated oncogene PIK3CA of this pathway encodes the catalytic subunit of PI3K [45]. Tumors with PIK3CA mutations respond strongly to PI3K pathway inhibitors alone and in combination with IR, representing a promising target for treatment [34,44,46,47]. The PI3K pathway contains, among other components, the mechanistic target of rapamycin (mTOR) [48]. mTORC1 is a protein kinase and an upstream activator of ribosomal protein S6 kinase (S6) that regulates lipid synthesis, energy metabolism, autophagy, and lysosome function and maintains cell homeostasis [49], while mTORC2 prevents apoptosis [50], regulates metabolic control, and influences actin polarization [51]. In addition, mTOR-I is thought to enhance radiosensitivity by suppressing both HR and NHEJ, thereby inhibiting the repair of DNA DSBs induced by IR [52].

In our study, we focused on the three KIs AZD7648, Sapanisertib, and CC-115 combined with IR. AZD7648, which is a strong and selective DNA-PK-I, enhances the sensitivity of cells to radiation both in vitro and in vivo [53]. AZD7648 is currently in two phase I clinical trials, including monotherapy and combination therapy with other anti-cancer drugs (NCT03907969) or IR (NCT05116254). The second KI we tested was Sapanisertib, also known as TAK-228, MLNO128, or INK128, which inhibits both subcomplexes of mTOR (mTORC1 and mTORC2) and decreases the phosphorylation of mTOR [54]. Recent in vivo studies have shown promising results with Sapanisertib when used in combination therapy for nasopharyngeal carcinoma [55,56]. In other tumor entities, there have been studies in murine models with Sapanisertib as a radiosensitizer [54]. Several clinical trials are currently in phase I and II with Sapanisertib as a mono- and combination therapy. The third KI we tested was CC-115, a dual inhibitor of both mTOR and DNA-PK. CC-115 inhibits the auto-phosphorylation of the catalytic subunit of DNA-PK (DNA-PKcs) at the S2056 site (p-DNA-PK S2056) and prevents its dissociation from DNA ends. Moreover, CC-115 has high kinase selectivity and is a strong inductor of apoptosis in solid tumor cancer cell lines [57]. Clinical trials with CC-115 in phase I (NCT01353625) and phase II (NCT02977780) are currently underway.

Combined treatment with kinase inhibitors and ionizing radiation (KI + IR) has the potential to not only eradicate tumor cells but also induce damage to surrounding normal tissue, leading to more severe side effects. Therefore, we examined how the treatment affected both HNSCC and healthy tissue cells [58]. We studied various aspects, including cell proliferation, senescence, cell death, the distribution of cells in the cell cycle, and their influence on the mTOR signaling pathway.

## 2. Materials and Methods

### 2.1. Cell Cultures and Inhibitors

The HNSCC cell lines CAL33, UD-SCC-2, UM-SCC-47, and HSC4 were provided by Dr. Thorsten Rieckmann (University Medical Center Hamburg-Eppendorf, Germany) and cultivated in Dulbecco’s MEM (DMEM; Gibco, Waltham, MA, USA) supplemented with 10% FBS (Merck, Darmstadt, Germany) and 1% penicillin/streptomycin (Gibco, Waltham, MA, USA). CAL33 and HSC4 are HPV^−^, while UM-SCC-47 and UD-SCC-2 are HPV^+^. SBLF7 and SBLF9 are normal human skin fibroblasts and were isolated from healthy donors by skin biopsy and subsequently cultivated with 15% FBS; 2% non-essential amino acids (NEA; Merck, Darmstadt, Germany); and 1% penicillin/streptomycin in F-12 medium (Gibco, Waltham, MA, USA). HaCaT are healthy human skin keratinocytes and were cultivated in Dulbecco’s Modified Eagle Medium with a glucose concentration of 4.5 g/L (DMEM (1X) + GlutaMAX − I; Gibco, Waltham, MA, USA) supplemented with 10% FBS and 1% penicillin/streptomycin. All cells were incubated at 37 °C in a humidified 5% CO_2_ atmosphere.

CC-115, AZD7648, and Sapanisertib (Selleckchem, Houston, TX, USA) were diluted in dimethyl sulfoxide (DMSO, Roth, Karlsruhe, Germany) and stored at −80 °C. The selected concentrations of KI were determined via a dilution series using CAL33 cells. Required aliquots were freshly thawed prior to each experiment.

### 2.2. Determination of Dose

To find the appropriate concentration of the three inhibitors, CAL33, HSC4, and UM-SCC-47 were treated using different concentrations with and without IR and then analyzed via a colony-forming assay and the flow-cytometric determination of cell death. For further experiments, concentrations of 1 µM for CC-115, 5 µM for AZD7648, and 0.5 µM or 1 µM for Sapanisertib were chosen.

### 2.3. Colony-Forming Assay

Cells were seeded in 60 mm Petri dishes (Thermo Fisher Scientific, Roskilde, Denmark) at a density of 720 to 7200 cells per dish and incubated for 24 h at 37 °C before the addition of the KIs CC115 (1 μmol/L), AZD7648 (5 μmol/L), or Sapanisertib (0.5 μmol/L). We delivered 2 Gy IR 3 h later by an ISOVOLT Titan X-ray generator (GE, Ahrensburg, Germany). After 24 h of incubation, the medium containing the drug was replaced with fresh medium. Cultures were incubated for 1–3 weeks until colonies were formed. Colonies were stained with methylene blue (#66725, Sigma Aldrich, Munich, Germany) for 30 min at room temperature. Clusters containing 50 or more cells were considered to be a colony. Image analysis software (Biogas Kobi 1.2) was used to count the colonies (Biomas, Erlangen, Germany). The determination of plating efficiency (PE) involved calculating the ratio of the colonies formed to the initially seeded cells. The survival fraction (SF) was then computed by dividing the number of colonies formed by the PE and multiplying the result by the number of cells initially seeded [59].

### 2.4. Immunohistochemical Staining

Cells were cultured on cover slips to 90% confluence maximum and incubated with the KIs CC115 (1 µmol/L), AZD7648 (5 µmol/L), or Sapanisertib (0.5 µmol/L) for 24 h. Three hours after the application of the drugs, the cells were irradiated with 2 Gy. After 2, 4, 8, 16, or 24 h of recovery, cells were fixed with 4% formaldehyde and blocked with BSA overnight. The stained targets were γH2AX, Ki-67, p21, α-Tubulin, mTOR, DNA-PKcs, mitochondria, promyelocytic leukemia protein (PML), phosphorylated-DNA-PKcs (p-DNA-PKcs), S6, phosphorylated-mTOR (p-mTOR), and phosphorylated-S6 (p-S6). The staining with specific antibodies was carried out overnight at 8 °C. The secondary antibodies AlexaFluor488 goat anti-mouse, AlexaFluor 647 goat anti-rat IgG, and AlexaFluor555 anti-rabbit (1:2000, Invitrogen^TM^, Waltham, MA, USA) were incubated for 1 h at room temperature. Cover slips were transferred onto glass slides using Vectashield (Vector Laboratories, Burlingame, CA, USA). Gray-scale images were acquired using a fluorescence microscope (Axioplan 2, Zeiss, Göttingen, Germany) and imaging software (Metafer 4, MetaSystems, Altlußheim, Germany).

### 2.5. Apoptosis and Necrosis Analysis by Flow Cytometry

Cells were seeded in T25 flasks at a density of 150,000–200,000 cells per flask and then incubated at 37 °C for 24 h before the addition of the KIs CC-115 (1 μmol/L), AZD7648 (5 μmol/L), or Sapanisertib (1 μmol/L) in serum-reduced medium containing only 2% FBS. The cell cycle was not halted by this serum concentration, and the cells passed through it normally [60,61,62]. After 3 h, the cells underwent irradiation with 2 Gy. Following 48 h of treatment, both the cells and supernatant were harvested and stained with a mixture of 7 AAD and APC Annexin V (BD Biosciences, Franklin Lakes, NJ, USA) in equal parts. The staining process occurred on ice for 30 min, after which the dye was removed, and cells were resuspended in cold Ringer’s solution (Fresenius, Bad Homburg, Germany). Flow cytometry analysis was performed using a Cytoflex flow cytometer (Cytoflex, Beckman Coulter, Brea, CA, USA), and data analysis was performed using Kaluza analysis software (Kaluza 2.1, Beckman Coulter, Brea, CA, USA).

### 2.6. Cell Cycle Analysis by Flow Cytometry

Cells were seeded, treated, and harvested following the same procedures as those employed for cell death analysis. Subsequent to harvesting, fixation was carried out using 70% ethanol and 2% FBS medium, and the samples were stored at 5 °C for at least 12 h. Hoechst 33258 staining (Molecular Probes, Eugene, OR, USA) was then applied, with an incubation period of 60 min on ice. Following staining agent removal, cells underwent centrifugation and were resuspended in cold Ringer’s solution. Flow cytometry and Kaluza analysis software (Kaluza 2.1) were employed to analyze the distribution across various cell cycle phases.

### 2.7. Live-Cell Microscope (Real-Time Microscope)

Cells were seeded in 24-Well Cell Culture Plates (Greiner bio-one, Frickenhausen, Germany) with a density of 19,000 cells per well and placed in the incubator at 37 °C under the 24-channel real-time microscope (zenCELL owl, Espelkamp, Germany) with an acquisition frequency of 1 h. Twenty-four hours later, the KI was administered, and the camera was promptly started. IR of 2 Gy was delivered 3 h later. After an incubation period of 24 h, the medium containing the drug was replaced. The cultures were further incubated and monitored until complete overgrowth occurred across the entire surface of the cell culture plate, at which point the camera recording was terminated. Cell numbers over time were calculated by ZenCell Owl software (ZenCellOwl 3.4.1). Cell numbers were fitted by an exponential growth equation, y=y0×exp⁡(k×X), where *y*0 is the initial cell count, *k* is the growth rate, and *X* is the time interval. The doubling time was calculated as follows: doubling time=ln⁡(2)/k.

### 2.8. Statistical Analysis

For the statistical analysis and graphs, IBM SPSS Statistics 28 (IBM, Armonk, NY, USA) and GraphPad Prism 8 software (GraphPad Software, San Diego, CA, USA) were used. Cell migration and cell death data were analyzed using an unpaired, one-tailed Mann–Whitney U test, as well as cell cycle data from the DNA-PK-I monotherapy. For cells treated with mTOR-Kinase-Inhibitor (mTOR-I), cell cycle data were evaluated using an unpaired, two-tailed Mann–Whitney U-test. A *p*-value ≤ 0.05 was determined as significant.

## 3. Results

In this study, we explored the influence of a combined treatment regimen involving IR, mTOR-I, and DNA-PK-I on four distinct HNSCC cell lines as well as normal tissue (Figure 1). The inhibitors utilized in this study were the DNA-PK-I AZD7648, the mTOR-I Sapanisertib, and the dual mTOR-I/DNA-PK-I CC-115. The immediate and long-term impacts of our treatment were scrutinized through cell death analysis, cell cycle analysis, real-time microscopy, and CFA, while the underlying molecular mechanisms were elucidated via immunohistochemical staining (IS).

### 3.1. γH2AX Immmunostaining to Detect DSBs

To study the immediate effect of the three inhibitors and IR on HNSCC and normal cells, an IS-based assay from 0 to 24 h was performed. Cell damage induced by IR can be analyzed by γH2AX, which marks the DSBs (Figure 2A). DNA-PK-I inhibits the repair of these DSBs, leading to a delayed or prolonged recovery. For this purpose, the induced foci per cell were counted, and their average numbers were compared over time (Figure 2B). We noted a rapid increase in γH2AX foci after IR in the UM-SCC-47 cell line, reaching a peak at 4 h and a decline to baseline levels by 16h. The IR + mTOR-I treatment led to a one-third higher average number of γH2AX foci per cell than in the control group after 4h. Within the observed time period, the number of γH2AX foci was elevated under IR + DNA-PK-I treatment compared to the control group, implying a slower decrease rate. The treatment with the combined inhibitor caused a lower initial rise in γH2AX foci compared to the control, yet it maintained a constant elevation throughout the duration of the experiment.

Next, the γH2AX foci per cell 24 h after IR in different HNSCC and normal tissue cells were counted, and the numbers were compared between the control and the KI-treated group (Figure 2C,D). At 24 h post-irradiation, normal tissue cell lines exhibited a lower count of γH2AX foci compared to HNSCC cell lines. Furthermore, the combination treatment of IR + Sapanisertib resulted in HaCaT keratinocytes in a reduced number of γH2AX foci compared to irradiation alone. The highest number of foci remained in the IR + AZD7648-treated group across all HNSCC cell lines. Therefore, we were interested in the extent to which cell death increased.

### 3.2. Assay for Apoptosis and Necrosis by Flow Cytometry

To detect cells undergoing apoptosis or necrosis shortly after the 24 h of treatment, flow cytometry was performed after staining the samples with Annexin-V APC and 7 AAD (Figure 3A). Annexin-stained cells were defined as “apoptotic”, 7AAD- and Annexin-stained cells as “necrotic”, and unstained cells were defined as “alive”. Our cell death analysis revealed a clear discrepancy in the response to KI + IR between normal tissue cells and HNSCC (Figure 3B). Compared to fibroblasts, a larger proportion of tumor cells entered apoptosis and necrosis. In all HNSCC cell lines, the co-administration of KI + IR, along with either CC-115 or AZD7648, resulted in a distinctly higher percentage of dead cells compared to irradiation alone. HNSCC cells treated with CC-115 + IR exhibited an elevation ranging from 59 to 121%, contingent on the specific cell line, in comparison to IR treatment alone. Similarly, AZD7648-treated HNSCC cells displayed an increase in cell death ranging from 25 to 150%. Notably, among CC-115-treated cells, necrotic cells significantly predominated over apoptotic cells, while AZD7648-treated cells exhibited a roughly equal distribution of necrotic and apoptotic cells. On the other hand, KI + IR with the selective mTOR inhibitor Sapanisertib demonstrated a significant response only in the HPV^+^ cell lines UM-SCC-47 and UD-SCC-2. In these instances, the apoptotic portion surpassed the necrotic portion. Next, we were interested in whether the cell cycle was arrested by the treatment in the radiosensitive G2/M phase.

### 3.3. Cell Cycle Analysis by Flow Cytometry

The impact of the treatment on the cell cycle distribution was determined by flow cytometric analysis after staining with Hoechst 33258. An increase in cells in G2/M arrest after 2 Gy irradiation and especially after the combined treatment with AZD7648 and IR was evident (Figure 4A). All tumor cells examined showed a higher increase in cells in the radiosensitive G2/M phase than healthy fibroblasts (Figure 4B). No increase in cells in the G2/M phase was seen when cells were treated with CC-115, AZD7648, or Sapanisertib alone. Compared to healthy fibroblasts, KI + IR resulted in an increased population in the G2/M phase in HNSCC cells. The specific DNA-PK-I AZD7648 in combination with IR led to a clear accumulation of G2/M arrest in most HNSCC cells, whereas this effect from IR ± CC-115 or Sapanisertib was only seen in HPV^+^ cells. Another important point is the inhibition of proliferation by IR + KI, so the next step was to look at the growth rate.

### 3.4. Real-Time Microscopy to Detect Time-Resolved Proliferation under Treatment

The medium-term effects of the KIs and IR, such as cell proliferation, were investigated by studying HNSCC cells and healthy tissue cells until the surface of the cell culture plate was overgrown. For this purpose, images were acquired hourly under the continuous incubation of the cells until a cell lawn was formed in the control group (Figure 5A). The cell number and coverage in the images was determined by Zen Cell microscope software. For further evaluation, only phenotypically normal-shaped “flat cells” were considered. The number of flat cells was determined and plotted over time, and the data were used for a non-linear regression to generate a growth curve. Approximately 3 days after treatment, the cell culture plate was overgrown. The KI-treated but not IR-exposed HNSCC cells like CAL33 displayed a delayed growth curve compared to the untreated control group. Therefore, growth was slowed down the least under CC-115 and the most under AZD7648. This effect was amplified multiple times by additional IR; the growth curve of all three KI ± IR-treated cell groups flattened strongly compared to the IR-exposed cell groups (Figure 5B). The resulting doubling times could be used to estimate the long-term proliferation of the cells. All three combination therapies resulted in distinct increases in doubling time compared to the control in CAL33 cells, except for the control group (Figure 5C). The increase in doubling time by adding KIs before IR exposure was also observed in other HNSCC and a normal tissue cell lines. Among the IR-exposed cell groups, a longer doubling time was shown in 11 of 12 groups by adding KIs. Compared to the untreated group, monotherapy with one of the three KIs resulted in slight growth retardation in most cell groups (Figure 5D). The most important factor for the success of therapy is that the cells finally stop dividing; therefore, we performed CFA next to determine the effect on long-term growth.

### 3.5. Colony-Forming Assay for Clonogenicity

The CFA was performed to assess the impact of mechanisms such as cell survival, senescence, and clonogenicity (Figure 6). Additionally, the assay served to evaluate the effectiveness of drug-induced cytotoxicity and to observe potential synergistic effects resulting from combined IR and KI treatment. The concentrations of the inhibitors were selected so that the clonogenicity remained the same under monotherapy within a cell line. Among the three KIs, AZD7648 was most effective in all cell lines investigated. HPV^+^ and HPV^−^ HNSCC cells responded equally to AZD7648 + IR. The KIs CC-115 and Sapanisertib achieved synergistic reductions in clonogenicity only in HPV^+^ HNSCC. HaCaT keratinocytes showed higher clonogenicity with Sapanisertib ± IR than in the control group without kinase inhibitors. The synergistic reduction of the survival fraction was also achieved in HPV^−^ CAL33 tumor cells by CC-115 + IR. No synergistic effects were observed in the other groups. After analyzing the impact of the treatment on a cellular level, we extended our research to investigate the molecular effects of mTOR-I and IR on the mTOR signaling pathway.

### 3.6. Immunohistochemical Staining of the Downstream Proteins of mTOR

In addition to the mTOR inhibitors CC-115 and Sapanisertib, IR exerts an impact on the mTOR signaling cascade. Notably, IR not only directly activates mTOR but also fosters the proliferation of S6 [63]. To obtain a more complete understanding of the varied effects of the respective mTOR-Is + IR, we investigated their impact on the downstream protein S6 by measuring its levels of phosphorylation via the IS of p-S6 in UM-SCC-47 HNSCC tumor cells over a period of 0 to 16 h after IR exposure (Figure 7A) [64].

Following IR, the signal intensity of p-S6 showed a rapid increase, peaking at almost twice the initial intensity at 4 h. It then gradually declined, returning to baseline 8 h after irradiation and falling below the initial value by the 16 h mark. When CC-115 was added, the intensity of p-S6 decreased by more than a third and remained at a constant level throughout the study period, even with additional irradiation. In contrast, the addition of Sapanisertib had minimal effect on the amount of p-S6. However, when combined with additional irradiation, the signal intensity of p-S6 decreased rapidly, reaching a level similar to CC-115 + IR 2 h after IR, and continued to decrease over time. Consequently, the signal intensity remained consistently reduced by more than half of the original value between 4 and 16 h (Figure 7B).

## 4. Discussion

Currently, there is a great need for improved therapy alternatives for HNSCC, since most patients are diagnosed in advanced stages [2]. We focused our research on radiosensitizing DNA-PK-I and mTOR-I, which promise high therapeutic success in combination with ionizing radiation. Advanced HNSCC tumors often contain mutations in the PI3K–AKT–mTOR pathway and respond strongly to KIs such as the selective mTOR-I Sapanisertib, which we assessed in our study [44,55]. A further KI we studied was AZD7648, which is a specific DNA-PK-I and suppresses the DNA repair of IR-induced DSBs [53]. The dual KI CC-115 inhibits both mTOR and DNA-PK proteins [57].

The aim of our research was to assess the effectiveness of KI + IR treatment on tumors and the influence on surrounding cells. Therefore, we compared the effects of IR alone and IR + KI treatment by using IS, cell cycle analysis, apoptosis and necrosis analysis, real-time microscopy, and CFA. We also considered the HPV status of the tumor cells, since it is known that this affects the genetic background and reaction to IR.

γH2AX is linked to IR-induced DSBs and therefore can detect cell damage [65]. Since DNA-PK-I prevents DSB repair, the remaining γH2AX foci were evaluated in IS 24 h after IR. The highest number of γH2AX foci was detected in all AZD7648 + 2 Gy-treated samples of HNSCC, reflecting the highest amount of sustained DNA damage over 24 h. CC-115 and Sapanisertib ± IR also led to a higher number of DSBs than IR treatment alone, whereas KIs without IR did not induce any additional remaining DSBs 24 h after IR exposure. This demonstrated the radiosensitizing effect of the three KIs in HNSCC. In healthy fibroblasts, no visible cell damage remained after 24 h under any of the applied therapies, in contrast to keratinocytes, which responded to the treatment similarly to HNSCC cells.

In many HNSCC cells, repair pathways such as HR are impaired, while they are functional in normal tissues such as fibroblasts. When the NHEJ is blocked by DNA-PK-I, normal tissue cells can use HR, whereas this pathway is often disrupted in cancer cells. This may explain the increase in the cell death rate after IR + DNA-PK-I treatment [33,44,66]. In our study, necrotic and apoptotic cell rates were clearly increased in six of eight combination therapies with different HNSCC cell lines and DNA-PK-Is. Since HPV^+^ HNSCC cells are more sensitive to IR than HPV^−^ HNSCC cells, the higher cell death rate observed in HPV^+^ HNSCC cells can be attributed to this increased sensitivity. In contrast, the specific mTOR-I Sapanisertib led to a higher cell death rate than IR alone only in HPV^+^ HNSCC cells. Consequently, Sapanisertib did not augment IR sufficiently to bring about a significant change in cell death in the more IR-resistant HPV^−^ HNSCC cells.

The main effect of the KI + IR treatment goes beyond the induction of cell death. For radiosensitivity, the repair pathways of induced DNA damage as well as the cell cycle phase of the cell are essential. The most radiosensitive phase of the cell cycle is the G2/M phase, followed by the G1 phase. In contrast, the S phase is the most radioresistant in the cell cycle [43]. Conditions like DSBs or increased radiosensitivity can result in a G2 block in the cell cycle, which increases the number of cells in the G2/M phase. This can be approached therapeutically through IR or KIs to induce cell cycle arrest, increasing radiosensitivity. In addition to its function in DSB repair, DNA-PK is also involved in cell cycle regulation [34]. We used Höchst 33258 staining to investigate whether IR + KI led to an increased number of cells in the G2/M phase. However, normal tissue cells were not affected by KIs or KIs + IR, and none of the single-KI treatments had an impact on the cell cycle of HNSCC cells. The majority of HNSCC cells responded, with an increased population of cells in the G2/M phase under AZD7648 + KI treatment, while Sapanisertib or CC-115 + KI led mostly to G2 arrest in HPV^+^ HNSCC cells. However, our data suggest that neither mTOR-I nor DNA-PK-I when used alone had an influence on the cell cycle. In combination with IR, mTOR-I led to a G2 block only in HPV^+^ HNSCC cells, while DNA-PK-I induced G2 arrest in HPV^+^ and HPV^−^ HNSCC cells.

One option to measure the response to a treatment is through its effect on cell proliferation. Therefore, we calculated the doubling time of the different treatments by generating a growth curve using real-time microscopy. Due to the limitation of the observation period of 3 days after treatment, the effect on the long-term growth was not assessed, but short-term effects on cell proliferation could be observed. While the single-KI treatment showed low growth inhibition in most cell lines, the combination therapies were significantly more efficient, corresponding to the effects on cell death and cell cycle distribution.

Another important aspect in evaluating response to therapy is clonogenicity, including cell death and senescence. Conditions like cellular stress via DNA damage or ageing processes can lead to an irreversible growth arrest to prevent the persistency of mutated cells, which is called cellular senescence [67]. CFA was suitable for not only analyzing the long-term effects of the treatment, but also describing the additive or synergistic effects of the combination therapies. In our setting, we detected strong synergistic effects of simultaneous AZD7648 and IR therapy on the colony-forming capacity, which may be indicative of senescence, in all investigated cell lines. CC-115 and Sapanisertib in combination with IR achieved synergistic reductions in clonogenicity in only HPV^+^ HNSCC cells and one of four groups of HPV^−^ HNSCC cells. The apparent radioprotective effect of Sapanisertib on HaCaT keratinocytes was probably not due to the mTOR inhibitor, but rather to the increased number of cells during seeding.

According to our dose-finding study via CFA and flow cytometry, Sapanisertib is highly toxic as a monotherapy and induces cell death in HNSCC and normal tissue cells. For this reason, it is only applicable at low doses. In terms of cell proliferation, radiosensitivity, and cell death, a therapeutic effect could only be achieved in HPV^+^ HNSCC cells. In the case of the repeated use of Sapanisertib + IR in several therapy cycles, stronger success can be assumed. According to our assessment, this is not recommended due to the high toxicity of Sapanisertib. Consequently, targeted therapeutic approaches in the frequently mutated PI3K–AKT–mTOR pathway in HNSCC cells do not provide an advantage over RT, nor do they bring a supra-additive advantage to KI + IR, as the surrounding tissue is also sensitive to Sapanisertib.

On the other hand, based on our dose-finding study, the dual KI CC-115 in monotherapy was less toxic in terms of cell death for normal tissue cells than Sapanisertib, allowing slightly higher concentrations to be used for KI + IR therapy. This may also explain the stronger effect of CC-115 than the selective mTOR-I in terms of cell death and cell reproductive death.

Regarding mTOR signaling, we investigated the phosphorylation of S6 in HNSCC, a process regulated by mTOR [64]. Consistent with findings from prior studies, we observed an augmentation in mTOR activity following IR [68]. Specifically, our research revealed an initial increase in the signal of p-S6 in IS immediately after IR, demonstrating enhanced mTOR activity.

In response to ionizing radiation (IR), the activity of mTOR was slightly upregulated, persisting for approximately 8 h, followed by a subsequent attenuation in mTOR activity. However, our observations within the context of CFA (Figure 6) did not indicate a proliferative effect, despite the transient increase in mTOR activity induced by IR. Considering the high prevalence of DNA double-strand breaks during this period (Figure 2B), the increased mTOR activity appears to serve as a strategic mechanism to ensure cellular survival.

The dual mTOR-I/DNA-PK-I CC-115 led to the inhibition of mTOR activity, although this did not increase further after the addition of IR. The selective mTOR-I Sapanisertib showed only a minor inhibition of mTOR activity. However, when combined with IR, mTOR activity was more strongly suppressed than under CC-115 + IR or IR alone. Consequently, simultaneous tumor therapy with Sapanisertib + IR in the irradiation field would lead to strong mTOR inhibition and hardly affect mTOR activity in the surrounding healthy tissue, whereas CC-115 would lead to a decrease in mTOR activity throughout the body and thus also impair processes such as cell proliferation and cell differentiation in healthy body cells. Furthermore, the addition of IR had no effect on mTOR inhibition by CC-115.

In terms of human application, the potential use of the KIs investigated in this study holds great promise for addressing the challenges posed by advanced stages, therapy-resistant tumors, unresectable cases, and recurrent HNSCC. Given the current dissatisfaction with available therapeutic options in these scenarios, the consideration of the use of the KIs investigated in this study is particularly relevant. AZD7648 is a promising candidate for simultaneous RT in both HPV^+^ and often therapy-resistant HPV^−^ HNSCC. Of the KIs investigated in our study, AZD7648 showed the most promising in vivo results. CC-115 also appeared to be a viable option for use in both HPV^+^ and HPV^−^ HNSCC. In contrast, Sapanisertib showed a sufficient response only in HPV^+^ HNSCC, where successful responses to current established treatment options are more common than in HPV-negative cases.

## 5. Conclusions

In conclusion, our study provides insights into potential therapeutic strategies for HNSCC using targeted KIs ± IR, especially considering the high incidence of advanced-stage diagnoses and their limited therapeutic approaches and resistance to established treatment approaches. Considering the importance of sparing healthy surrounding tissue, it was found that targeting the DNA repair pathway proved to be more effective than targeting the frequently mutated PI3K–AKT–mTOR pathway in HNSCC. The DNA-PK-I AZD7648 + IR demonstrated a significant reduction in cell viability and proliferation in all HNSCC cell lines while sparing the normal tissue cell lines. This outcome suggests its promising potential as a radiosensitizer in tumor therapy. Particularly for HPV^−^ advanced-stage HNSCC, for which effective therapeutic options are currently lacking, AZD7648 + IR shows significant promise. The selective mTOR-I Sapanisertib primarily affected HPV^+^ HNSCC cells. However, its dosage was limited due to its high cellular toxicity in both HNSCC and healthy surrounding tissue cells. Combined mTOR/DNA-PK-I therapy with CC-115 did not provide a distinct advantage over the use of selective KIs in our investigations. However, the results regarding cell death and clonogenicity in HNSCC while sparing healthy tissue cells were superior to those observed under Sapanisertib. This implies limited benefits for its application in KI + IR therapy. Further research and clinical trials are essential to optimize the dosages and explore the full therapeutic potential of these kinase inhibitors in HNSCC treatment. In addition, animal experiments would provide an opportunity to more closely replicate the clinical setting.

## Figures and Tables

**Figure 1 cells-13-00304-f001:**
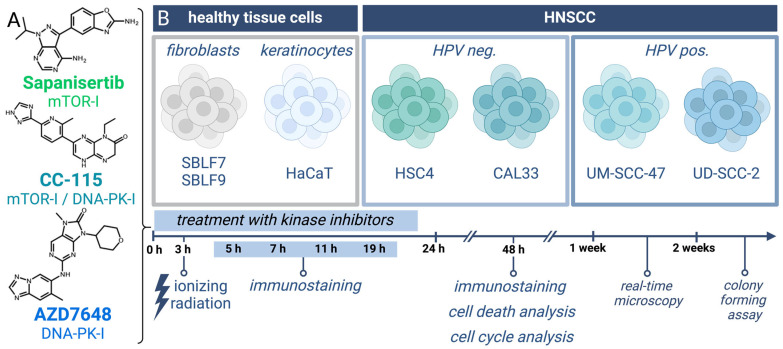
Treatment of HNSCC and normal tissue cells with KI + IR. (**A**) Chemical structures of the three kinase inhibitors: the selective mTOR-I Sapanisertib, the dual mTOR-I/DNA-PK-I CC-115, and the selective DNA-PK-I AZD7648. (**B**) Used cell lines and overview of treatment with endpoints of the different methods in a timeline (created with BioRender.com, accessed on 3 February 2024).

**Figure 2 cells-13-00304-f002:**
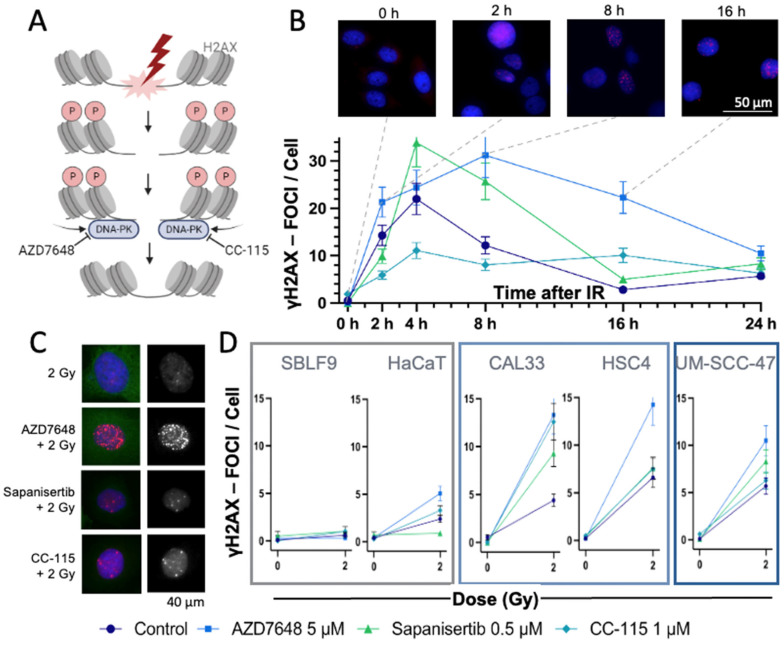
Analysis of DNA DSBs using γH2AX-IS. (**A**) IR leads to DNA DSBs, with DNA-PK involved in repair. When DNA DSBs occur, the histone protein H2AX is γ-phosphorylated and can be visualized by IS. DNA-PK-Is like AZD7648 or CC-115 block the repair of DSBs, thereby keeping H2AX phosphorylated and detectable as γH2AX foci via IS (created with BioRender.com, accessed on 12 December 2023). (**B**) Mean number of γH2AX foci in the nuclei of the HNSCC UM-SCC-47 cells over a period of 0 to 24 h after IR ± KI. Time 0 was not irradiated. Representative images of AZD7646-treated UM-SCC-47 stained with DAPI (blue) and γH2AX (red)—fluorescence microscopy image. Dashed lines indicate the times of the images with γH2AX foci. Error bars represent the SD of at least three replicates. (**C**) Exemplary presentation of fluorescence microscopy images of HNSCC CAL33 cells after IR ± KI. **Left**: The γH2AX foci are red and localized in the blue nuclei. **Right**: Monochromatic image of γH2AX foci. (**D**) Mean number of γH2AX foci in different normal (SBLF9, HaCaT) and HNSCC (CAL33, HSC4, UM-SCC-47) cell lines, 24 h after IR ± KI. Error bars represent the SD of at least three replicates.

**Figure 3 cells-13-00304-f003:**
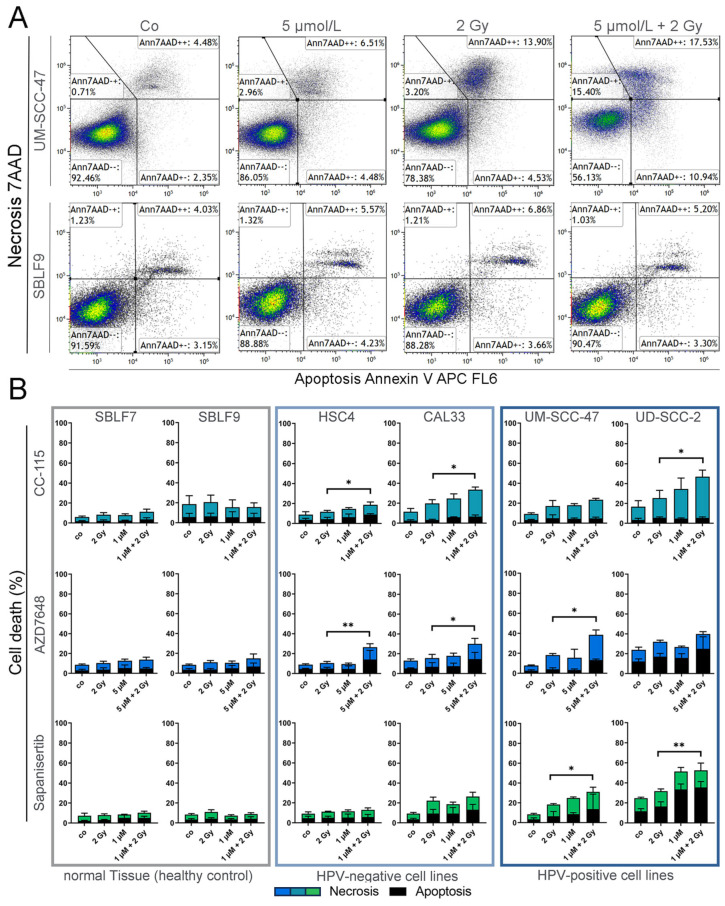
Flow-cytometric determination of cell death of HNSCC and healthy tissue cells. (**A**) Exemplary gating strategy of Annexin-V-APC/AAD staining. Dot plots of UM-SCC-47 (HNSCC) and SBLF9 (fibroblast) cells treated with 5 µmol/L AZD7648 (DNA-PK-I), 2 Gy irradiation, and combined IR and KI. (**B**) Cell death with necrotic and apoptotic components in normal tissue cells (SBLF7, SBLF9) and HPV^−^ (CAL33, HSC4) and HPV^+^ (UM-SCC-47, UD-SCC-2) HNSCC cells. Cells were treated with CC-115 (mTOR/DNA-PK-I), AZD7648 (DNA-PK-I), or Sapanisertib (mTOR-I); 2 Gy irradiation; or a combination of KI and IR. Each value represents mean ± SD (n ≥ 3). Significance was estimated by a one-tailed Mann–Whitney U test, * *p* ≤ 0.05 and ** *p* ≤ 0.01. *p* = 0.004: HSC4 + AZD7648, UD-SCC-2 + Sapanisertib; *p* = 0.0130: CAL33 + AZD7648; *p* = 0.0143: HSC4 + CC-115, CAL33 + CC-115, UD-SCC-2 + CC-115; *p* = 0.0283: UM-SCC-47 + Sapanisertib.

**Figure 4 cells-13-00304-f004:**
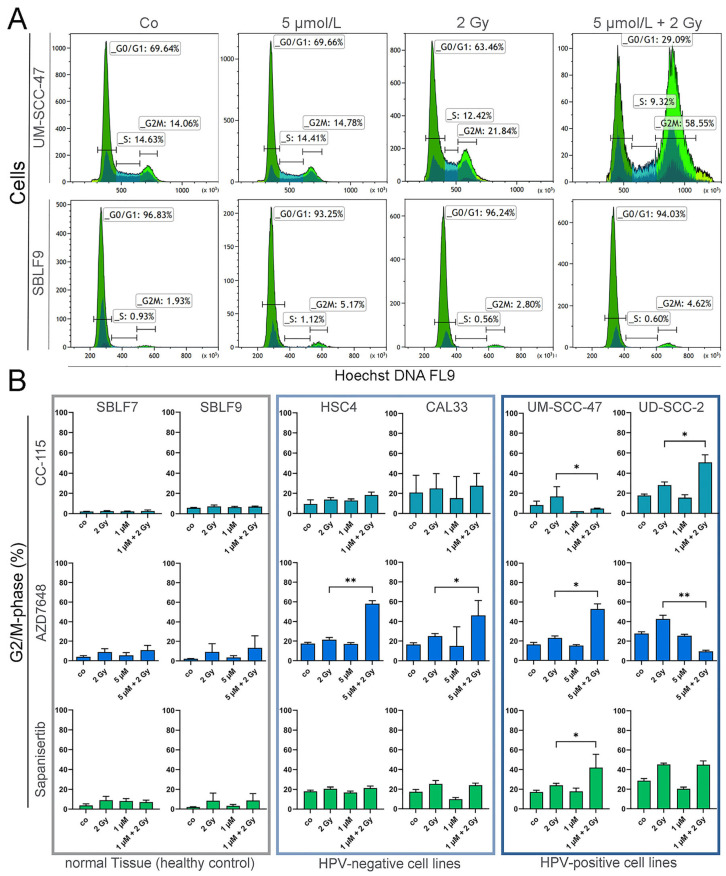
Cell cycle analysis of HNSCC and healthy tissue cells. (**A**) Representative histograms and gating strategy after treatment with 5 µmol/L AZD7648 (DNA-PK-I), IR with 2 Gy, and IR + KI treatment in UM-SCC-47 (HNSCC) and SBLF9 (fibroblast) cells. (**B**) Cells counted in G2/M phase after treatment with CC-115 (mTOR/DNA-PK-I), AZD7648 (DNA-PK-I), or Sapanisertib (mTOR-I); 2 Gy IR; or IR + KI. Cells are grouped into normal tissue cells (SBLF7, SBLF9) and HPV^−^ (CAL33, HSC4) or HPV^+^ (UM-SCC-47, UD-SCC-2) HNSCC cell lines. Each value represents mean ± SD (n ≥ 3). Significance was estimated by a one-tailed Mann–Whitney U test for mTOR-I-treated cells (CC-115 or Sapanisertib) and a two-tailed Mann–Whitney U test for DNA-PK-I as monotherapy (AZD7648), * *p* ≤ 0.05 and ** *p* ≤ 0.01. *p* = 0.004: HSC4 + AZD7648, UD-SCC-2 + AZD7648; *p* = 0.0143 UM-SCC-47 + AZD7648; *p* = 0.0238: UM-SCC-47 + CC-115; *p* = 0.0286: UD-SCC-2 + CC-115, UM-SCC-47 + Sapanisertib; *p* = 0.0325: CAL33 + AZD7648.

**Figure 5 cells-13-00304-f005:**
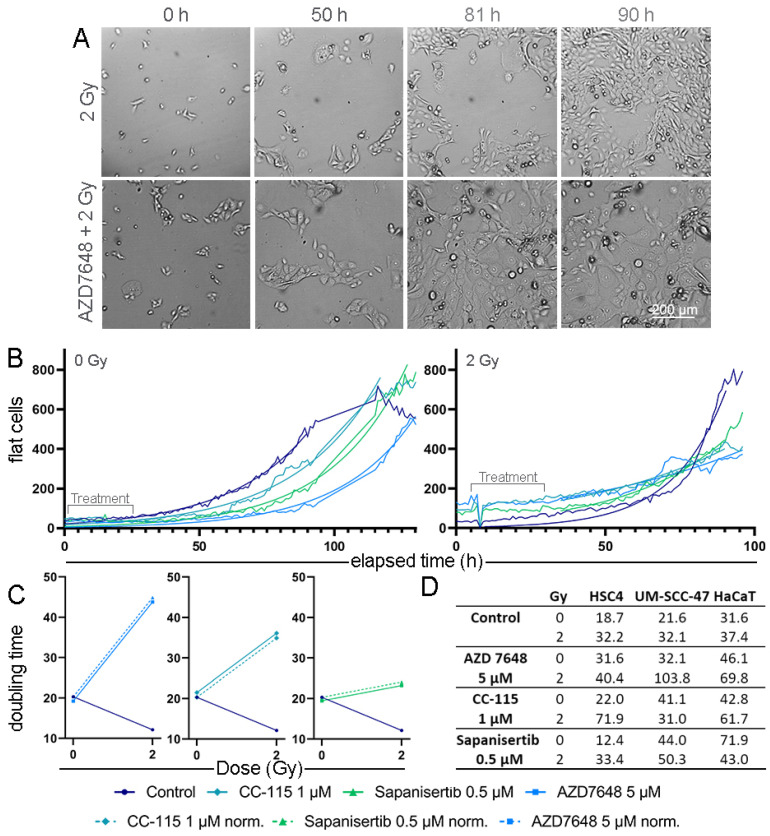
Real-time cell proliferation of HNSCC and normal cells under treatment. (**A**) Exemplary images of HNSCC CAL33 cells under IR ± 5 µM AZD7648 treatment starting from seeding to overgrowth of the surface. (**B**) Growth curve of normal-shaped CAL33 cells after IR ± KI treatment. An exponential growth equation was fitted from the data points. (**C**) Doubling time of CAL33 cells in hours under KI ± IR treatment, based on the exponential growth equation above. (**D**) Doubling time of HSC4, UM-SCC-47 (HNSCC), and HaCaT (healthy keratinocyte) cells in hours.

**Figure 6 cells-13-00304-f006:**
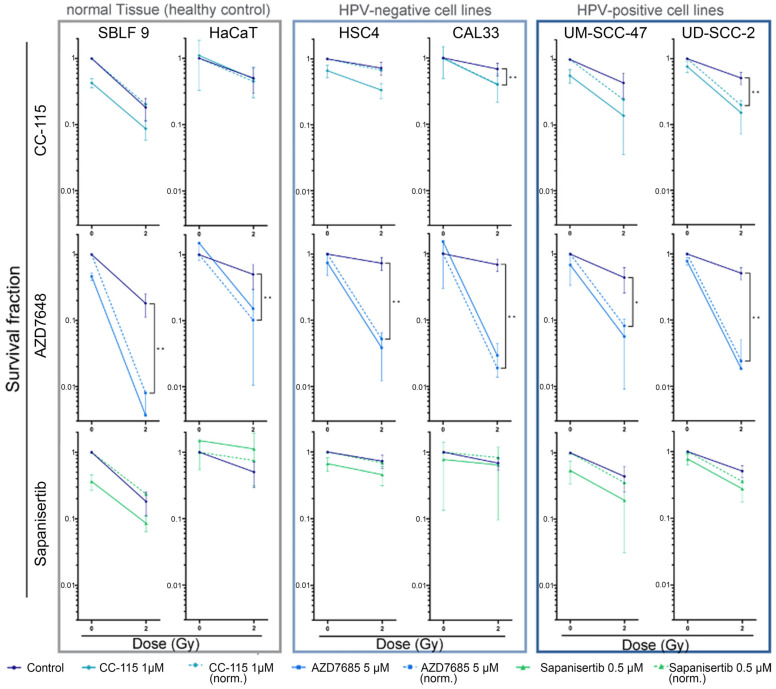
Colony-forming assay of HNSCC cell lines and normal controls. Cells are shown grouped into normal cells (SBLF9, HaCaT) and HPV^−^ (CAL33, HSC4) and HPV^+^ (UM SCC 47, UD SCC 2) HNSCC cells. Graphs show cell survival at 0 Gy and 2 Gy, either with additional treatment by KIs (1 µmol/L CC115, 5 µmol/L AZD7648, 0.5 µmol/L Sapanisertib) or without. Additionally, cell survival under KI treatment was normalized to detect synergistic effects. Dashed lines represent mean survival fraction normalized to the cytotoxicity induced by KI alone. Each value represents mean ± SD (n ≥ 3). Significance was estimated by a one-tailed Mann–Whitney U test, * *p* ≤ 0.05 and ** *p* ≤ 0.01. *p* = 0.001: UD-SCC 47 + CC-115 norm., SBLF9 + AZD7648 norm.; *p* = 0.009: HaCaT + AZD7648 norm., CAL33 + AZD7648 norm., HSC4 + Sapanisertib norm.; *p* = 0.018: CAL33 + CC115 norm.; *p* = 0.036: HSC4 + CC-115 norm.; *p* = 0.042: UM-SCC 47 + AZD7648 norm.

**Figure 7 cells-13-00304-f007:**
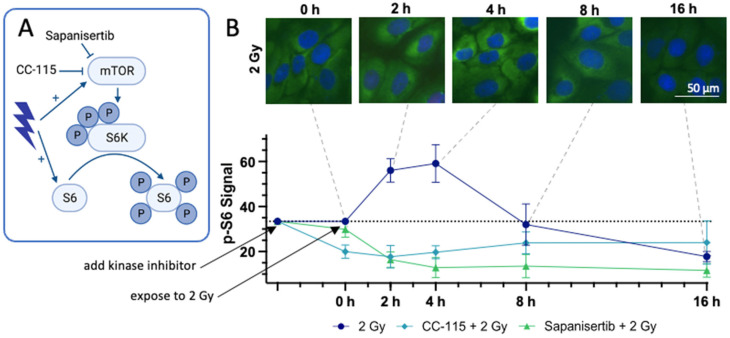
Effects of mTOR-Is + IR on the downstream protein S6 in WB. (**A**) Draft of the effect of IR, Sapanisertib, and CC-115 on the mTOR signaling cascade (created with BioRender.com, accessed on 16 January 2024). (**B**) Effects of CC115, Sapanisertib, and IR on mTOR signaling cascade in UM-SCC-47 tumor cells. The mean signal intensity of the phosphorylation of S6 (S 235/236) (green) was measured 0, 2, 4, 8, and 16 h after IR with 2 Gy via IS. The experiment was performed with the addition of CC-115, Sapanisertib, or no kinase inhibitor. The cell nuclei in the IS were stained blue using DAPI. Dashed lines indicate the times of the images with p-S6 intensity. Error bars represent the SD of at least three replicates.

## Data Availability

The datasets generated and/or analyzed in the present study are not publicly accessible, but can be provided by the corresponding author upon reasonable request.

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
