# Peer review of "Different Impacts of DNA-PK and mTOR Kinase Inhibitors in Combination with Ionizing Radiation on HNSCC and Normal Tissue Cells"

_cells, 2024, doi:10.3390/cells13040304_

Round 1

Reviewer 1 Report (Previous Reviewer 1)

Comments and Suggestions for Authors

Based on the responses I received from the authors, they have addressed all my queries and I have no further comments or concerns. I am satisfied with the manuscript now.

I suggest the manuscript to be published soon in your journal.

Reviewer 2 Report (Previous Reviewer 2)

Comments and Suggestions for Authors

The authors have improved the manuscript based on the comments.

This manuscript is a resubmission of an earlier submission. The following is a list of the peer review reports and author responses from that submission.

Round 1

Reviewer 1 Report

Comments and Suggestions for Authors

Good paper, some suggestions for clarification purposes.

Reviewer 2 Report

Comments and Suggestions for Authors

Different impacts of DNA-PK and mTOR kinase inhibitors in combination with ionizing radiation on HNSCC and normal tissues.

In this manuscript, Klieber et al. evaluated the effect of three kinase inhibitors, Sapanisertib, AZD7648, and CC-115, in combination with ionizing radiation on HNSCC and healthy tissue. Evaluating the efficacy and safety of new combination therapy strategies for HNSCC is very important for developing and bringing new treatment options to the clinic. This study is well-designed and has some significant results that could contribute to the development of new treatment options. I recommend that the authors consider the following recommendations and revise the manuscript before publication.

I appreciate that the authors included Figure 1, showing the treatment plan. This allows for an easy understanding of the experiment plan.

This is a general comment for all figures. I understand the authors used grey and blue backgrounds to differentiate between the types of cells used. However, using these colored backgrounds makes it difficult to clearly see the lines in the line plots (especially for someone like me, who finds it hard to distinguish between shades of blue and green). Please consider using a different color palette to make it easier to read the plots.

Figure 2. I am assuming that the control treatment shown in 2B is the cells treated with IR alone. It would be great if the authors could state that in the legend.

Figure 2. Please show the error bars in all plots and indicate significant differences. Also, mention how the significance was calculated and the number of repeats used in the legend.

Figure 2. Why do you think that treatment with CC-115 (which inhibits both DNA-PK and mTOR) triggered lower DSBs than the control group at 4 h?

Figure 3. In 3B, it is unclear whether the authors are comparing total cell death between the treatment groups or apoptotic cells and necrotic cells. Could you please revise the legends to make this clear?

Figure 3. Could the authors comment on the mechanism underlying the cell-type-specific differences in the treatment outcome?

Figure 4. Were the cells synchronized before the cell cycle analysis? The authors mentioned that the cells were serum-starved for 3 hours. Was this enough to synchronize the cells? It would be great if the authors could include a histogram showing cell synchronization at time 0 (maybe as a supplementary figure). Also, did the authors determine cell cycle progression at different time points after treatment? I think that to confirm that the cells were truly accumulating at a specific phase, they should be monitored at different time points, and the time taken to roll over from one phase to the next should be calculated.

In Figure 2D and Figure 6, it seems that treatment with Sapanisertib had a radio-protective effect in HaCaT. Could the authors comment on that?

Figure 6. I feel that the western blot images in this figure could be improved. In 6B, although the authors have plotted the band densities, I could not see clear differences between the band densities at different time points after treatment in the blot image. Also, could the author confirm that the plot showing the band densities was derived from replicates? Also, please indicate the error bars and significant differences in the plot.

Figure 7. I feel that the western blot images in 7A and especially in 7C could be improved. Please consider providing clearer blots so that the reader can see the trends shown in the band density plots in the blot images as well. Also, as mentioned in my previous comment, please include the error bars and show significance in the plots.

Minor comments

Please use the Greek letter β consistently in the manuscript.

γH2AX is written as yH2AX in some contexts. Please correct that.

Are the authors considering/planning an animal study based on these results? Please discuss.

Comments on the Quality of English Language

Some minor improvements/proofreading required.

Round 2

Reviewer 2 Report

Comments and Suggestions for Authors

Different Impacts of DNA-PK and mTOR Kinase Inhibitors in Combination with Ionizing Radiation on HNSCC and Normal Tissue Cells

I appreciate the authors for revising the figures. The new figures are much easier to read than the original ones.

The authors have addressed most of my concerns and have improved their manuscript. Please consider the following minor comments and revise the manuscript accordingly. Also, I recommend that the authors proofread the manuscript thoroughly and ensure that it is free of any language-related errors.

 Introduction. Line 38. HNCC ranks ‘amongst’  instead of ‘amount’. Alternatively, HNCC is one of the most …..

‘(Johnson)’ is mentioned 2 times in the Introduction. Not sure what this is. Please check.

Line 48: Please consider changing to “despite advancements” instead of “although advancements.”

Line 156: Please remove ‘.sg’ 

Comments on the Quality of English Language

Please proofread the manuscript thoroughly.